# Understanding the Postharvest Phytochemical Composition Fates of Packaged Watercress (*Nasturtium officinale* R. Br.) Grown in a Floating System and Treated with *Bacillus subtilis* as PGPR

**DOI:** 10.3390/plants11050589

**Published:** 2022-02-22

**Authors:** Giuseppe Pignata, Andrea Ertani, Manuela Casale, Diana Niñirola, Catalina Egea-Gilabert, Juan A. Fernández, Silvana Nicola

**Affiliations:** 1Department of Agricultural, Forest and Food Sciences, DISAFA University of Turin, UNITO, Via Leonardo da Vinci, 44-Largo Paolo Braccini, 2, 10095 Grugliasco, TO, Italy; giuseppe.pignata@unito.it (G.P.); manuela.casale@unito.it (M.C.); silvana.nicola@unito.it (S.N.); 2Department of Agronomical Engineering, Universidad Politécnica de Cartagena, Paseo Alfonso XIII, 48, 30203 Cartagena, Spain; dianicax@hotmail.com (D.N.); catalina.egea@upct.es (C.E.-G.); juan.fernandez@upct.es (J.A.F.); 3Brassica Breeding Group, Sakata Seed Ibérica S.L.U. Plaza Poeta Vicente García, 6, 46021 Valencia, Spain; 4Institute of Plant Biotechnology, Universidad Politécnica de Cartagena, Edificio I+D+i, Campus Muralla del Mar, 30202 Cartagena, Spain

**Keywords:** physiological changes, shelf life, postharvest, fresh-cut, baby leaf vegetables, soilless culture systems

## Abstract

The physiological changes and phytochemical pathways of processed watercress (*Nasturtium officinale* R. Br.) undergone during storage are not well known. The objective of this work was to evaluate the respiration rate and the inherent and external quality of watercress inoculated with *B. subtilis* and packaged as a fresh-cut product and stored at 4 °C for 11 days. Watercress was grown using continuous flotation (FL) in a greenhouse using substrate disinfection and inoculated or not with *Bacillus subtilis* as a plant-growth-promoting rhizobacteria (PGPR). The fresh-cut watercress respiration rate and phytochemical profile changed during the shelf life. The inherent phytochemical compounds were retained during the storage of the fresh-cut salad bags. The best results were found in watercress grown in a disinfected substrate but were less satisfactory when seeds and substrates were inoculated with PGPR. In general, the external quality and the pigment contents progressively decreased during the shelf life and the browning enzyme activities responsible for phenolic oxidation increased at different intensities throughout storage. At the end of the shelf-life period, the fresh weight loss of the fresh-cut product was less than 1% of the original weight. The results demonstrated that watercress grown in FL is a standardised baby leaf vegetable that is suitable for processing in the fresh-cut industry and for storing for more than 10 days. Unclear results were obtained for *Bacillus subtilis* in the postharvest period due to the inconsistent responses of the different analysed parameters.

## 1. Introduction

Watercress (*Nasturtium officinale* R. Br.) is a baby leaf vegetable (BLV) that is extensively consumed in Northern Europe, in particular as a fresh-cut product and in salad mixes [1,2,3]. Watercress is rich in carotenoids, antioxidants, phenols, vitamin C, phosphorus and calcium [4], and it has been used as a medicinal and food crop for over 2000 years [5]. Like any perishable product, fresh-cut leafy vegetables suffer from irreversible quality deterioration because the raw material quality can only be retained and not improved along the supply chain [6]. Chlorophyll *a* and *b* contribute to the green intensity and the external appearance of plant leaves, but they can degrade during the postharvest shelf life of fresh-cut products [7,8]. The yellowing that results from chlorophyll degradation is considered a common disorder for most leafy green vegetables during storage [9]. Similarly, another postharvest decay index is leaf tissue browning due to physiological dynamics, which involves phenolic oxidation and browning enzyme activity [10,11]. The antioxidant capacity and vitamin C content of BLVs could reduce the incidence of browning phenomena [12].

Among the phytochemical molecules, carotenoids and phenols are functional compounds that are degraded to different extents and re-biosynthesised in the BLV tissue [13,14,15]. In order to prolong the fresh-cut product shelf life, it is necessary to have a raw material with a high processing suitability and specific quality characteristics [16,17,18]. Currently, the fresh-cut vegetable shelf life is ca. 7 days in many European countries because it is affected by raw material quality, handling procedures, processing operations and storage conditions [6,19]. BLVs offer some advantages that make them particularly appreciated by the fresh-cut industry. The positive characteristics are related to its: (a) high percentage of edible product, which increases processing efficiency; (b) small whole leaves, which are suitable for mild and fast processing; (c) absence of cutting operations during postharvest, which results in the oxidation phenomena related to phenolic oxidation being reduced to a minimum; (d) attractive presentation and external quality; and (e) excellent phytochemical composition [20,21,22].

Soilless cultivation systems (SCS) in protected cultivations can be considered a valid alternative to traditional culture systems in soil [23,24]. Among the various SCS, floating growing systems (FGS) can be implemented with a continuous flotation (FL), in which trays continuously float on a water-bed or hydroponic nutrient solution [25]. The nutritional status of BLVs can be rapidly influenced by FL, therefore leading to different results suitable for postharvest storage [26]. 

Different microbial-based approaches, in the form of biostimulants, are now being proposed for enhancing crop yield [27]. Plant-growth-promoting rhizobacteria (PGPR) represent hopeful and sustainable solutions to increase plant growth and yield [28]. They have been shown to have stimulatory effects on the growth and yield of radish, potato, tomato, beans and ornamental plants [29], as well as to increase the antioxidant capacity in watercress [4]. Moreover, PGPR have the capacity to counteract several stressors in plants [30,31,32]. In fact, PGPR can increase plant tolerance to stress conditions through chemical and physiological changes that are identified as induced systemic tolerance [31]. 

There are many PGPR strains, such as *Azotobacter*, *Azospirillum*, *Bacillus*, *Rhizobium*, *Pseudomonas* and *Serratia*, which can be used to improve plant growth [33] through the generation of antioxidants, hormones, and vitamins [34]. Among PGPR, *Bacillus* species can form long-living spores and metabolites that stimulate plant growth and prevent pathogen infection [35]. The colonization of the roots by bacteria provides a nutrient source, and in exchange, plants are the recipient of bacterial products that stimulate plant growth and provide stress protection to their hosts [35]. In addition, some microbial agents, such as *Bacillus subtilis*, produce enzymes that degrade the precursors of some hormones related to senescence, helping plants to overcome stress situations [34].

Intense research efforts have been devoted to the development of antagonistic microorganisms to control postharvest diseases [36,37,38]. So far, biological controls of dry and soft rots with different biocontrol agents such as fungi, bacteria, and yeasts have been reported as effective under experimental conditions [39]. In modern horticulture, substrate disinfection and PGPR inoculation should favour the production of raw materials adapted to the fresh-cut sector, with limited or no use of plant protection products [4]. Little is known about the physiological evolution and phytochemical pathway of processed watercress during storage. 

The limited information available on fresh-cut watercress shelf life indicates that watercress leaves are highly perishable and have a 7-day storage [40]. The objective of this research was to study the effect of the inoculation of a watercress growth substrate with *Bacillus subtilis* through the evaluation of the respiration rate, inherent and external quality, and phytochemical composition of watercress packaged as a fresh-cut product and stored at 4 °C for 11 days. 

## 2. Materials and Methods

### 2.1. Plant Material

The raw material was obtained from cultivations carried out in the Experimental Centre of the Department of Agricultural, Forest and Food Sciences (DISAFA) (44°53′11.67″ N; 7°41′7.00″ E-231 m a.s.l.) in Tetti Frati, Carmagnola (TO), Italy, from June to July in an automatically controlled temperature greenhouse. All plants of watercress (*Nasturtium officinale* R. Br.) cv. Aqua Large Leaf (Tozer Seeds Co., Cobham, UK) were grown in a lab-scale pilot plant for FGS [41] equipped with 3 benches, each one split into 4 separated flotation beds filled with a nutrient solution. Each bench (replicate) accommodated 4 treatments. Each treatment consisted of 12 trays containing 300 plants each (ca. 1961 plants·m^−2^). 

The experiment comprised growing plants in 60-cell Styrofoam trays (0.51 × 0.30 m with cells of 0.044 m upper and 0.025 m lower diameters, respectively) containing a specific peat-based horticultural medium (Neuhaus Huminsubstrat N17; Klasmann-Deilmann^®^ GmbH, Geeste, Groß-Hesepe, Germany) floating in a NS. The sown trays were placed in a plastic greenhouse until seed germination. Four days after sowing, the trays were moved into a lab-scale pilot plant equipped with 3 benches, each one split into 4 separated flotation beds (2.50 × 1.40 m; 0.15 m depth) and filled with 200 L of a 40/60 N-NO^3−^/N-NH^4+^NS composed of (all in mM·L^−1^): 12 N, 2 P, 6 K, 2 Mg, and 2.5 Ca. Then, a Lysodin^®^ Multimix formulation of microelements (Intrachem Production S.r.l., Grassobbio, Bergamo, Italy) was added to the NS at a dose of 0.30 g·L^−1^.

The harvesting of treated and untreated plants with *B. subtilis* took place after 24 days of cultivation, and all plants were used for the further sampling. At harvest, the raw material was immediately transferred to the postharvest laboratory to be processed as fresh-cut product.

### 2.2. Bacterial Strain and Inoculation

A total of 50% of the substrate, which consisted of a specific peat-based horticultural medium (Neuhaus Huminsubstrat N17; Klasmann-Deilmann, GmbH, Geeste, Groß-Hesepe, Germany) was disinfected (DS) with a steam stream at 100 °C for 45 min. The remaining part was not disinfected (NDS). The commercial product Larminar^®^ (10^12^ CFU·g^−1^ of *B. subtilis* strain AP-01; Agrimor, Agricultura Moderna S.A., Madrid, Spain) was used as a bacterial inoculant (BI). The seeds were sterilised in 20% NaOCl (*w*/*v*) and washed three times with sterile deionised water. Inoculation was performed twice: the first inoculation was performed one day before sowing, and for the second, 50% of the disinfected substrate (DS) and 50% of non-disinfected substrate (NDS) were inoculated with Larminar^®^ at a dose of 0.5 kg·m^−3^.

A total of 50% of the disinfected seeds were inoculated by soaking for 1 h in a *B. subtilis* suspension at a concentration of 10^8^ CFU·mL^−1^ in 0.9% of a NaOCl solution (*w*/*v*) obtained from Larminar^®^ in Plate Count Agar (PCA) (Fluka Analytical, Sigma-Aldrich S.r.l., Milan, Italy). The non-inoculated (NBI) seeds were kept for 1 h in 0.9% NaOCl. Eleven days after sowing, re-inoculation was performed by placing the inoculated trays (substrate and seeds) on a solution containing 0.167% Larminar^®^/water (*w*/*v*). Shoots of twelve plants per treatment and per block were used for phytochemical analyses. During the postharvest period, six bags per treatment per block were used; we examined one bag per treatment per block at each sampling date throughout the shelf life. 

All the analytical determinations were carried out after 1, 3, 5, 7, and 9 days of storage (d1, d3, d5, d7, and d9, respectively) and at the end of the shelf life (d11) in the DISAFA analytical laboratories.

### 2.3. Processing, Packaging and Storage Conditions

The raw material was sorted in a cold temperature room, and any damaged and yellowing shoots were discarded. Then, 125 g of watercress were packaged in 0.25 × 0.35 m thermo-sealed bags that had previously been prepared with polypropylene film with the following characteristics: an oxygen permeance of 1990 cm^3^ m^−2^ d^−1^ bar^−1^, a carbon dioxide permeance of 7800 cm^3^ m^−2^ d^−1^ bar^−1^, a water vapour permeance of 5.8 g m^−2^ d^−1^ bar^−1^. a film thickness of 20 µm, and a weight of 18.2 g m^−2^ (Alvapack S.r.l., Bologna, Italy). The packaged samples were stored at 4 °C for an 11-day shelf life in refrigerated chambers (MEDIKA 600; C.F. di Ciro Fiocchetti & C. S.n.c., Luzzara, (RE), Italy) without light in the display cabinet. 

### 2.4. Bag Headspace Gas Analysis

The headspace gas composition in the fresh-cut salad bags (O_2_% and CO_2_%) was measured using a Check Point Handheld Gas Analyzer (PBI-Dansensor AS, Ringsted, Denmark). To avoid modifications in the headspace gas composition due to gas sampling, each bag was used only once [8].

### 2.5. Fresh-Cut Product Fresh Weight Loss

Fresh-cut product fresh weight loss (FWL) was measured by weighing the bags daily during storage and progressively calculated based on the value from d0 as a decay index of freshness. 

### 2.6. Pigment Content Analysis

Frozen tissue (1.0 g) from each sample was homogenised with 20.0 mL of an 80% acetone/water (*v*/*v*) solution and extracted at 4 °C for 12 h in the dark. The extract was filtered and used for the spectrophotometric determination of chlorophyll *a*, chlorophyll *b*, and carotenoid (Chl. *a*, Chl. *b* and *Car.*, respectively) contents at wavelengths of 662, 645, and 470 nm, respectively [42]. The results were expressed according to the Lichtenthaler and Wellburn (1983) formulas.

### 2.7. Antioxidant Capacity

Frozen tissue (2.0 g) from each sample was mixed with 20.0 mL of pure methanol, incubated for 60 min, homogenised, and subsequently centrifuged at 958× *g* at 4 °C for 15 min. The antioxidant capacity (AC) was determined using the ferric-reducing ability of the plasma (FRAP) assay [43], by mixing 30 µL of the sample methanol extract with 900 µL of the FRAP reagent [14]. The mixture was incubated at 20 °C for 4 min, and the absorbance was spectrophotometrically determined at a wavelength of 593 nm. 

### 2.8. Total Phenolic Content

The total phenolic (TP) content was determined by mixing 100 µL of the sample methanol extract, prepared as described in Section 2.6, and the antioxidant capacity by using the Folin–Ciocalteau reagent [44]. After incubating the mixture for 3 min, 400 µL of a 7.5% sodium carbonate/water (*w*/*v*) solution were added. The new solution was mixed, incubated at 20 °C for 30 min in the dark, and then used for spectrophotometric determination at a wavelength of 760 nm [42].

### 2.9. Browning Potential and Soluble o-Quinone Content

Frozen tissue (5.0 g) from each sample was homogenised with 10.0 mL of methanol and subsequently filtered with gauze and centrifuged at 15,040× *g* at 4 °C for 15 min [45]. The absorbance was spectrophotometrically determined at a wavelength of 340 nm to establish the browning potential (BP) and a wavelength of 437 nm to establish the soluble o-quinone (So-Q) content [46].

### 2.10. Enzyme Activity Analysis

Frozen tissue (0.5 g) from each sample was ground in a mortar with liquid N, mixed with 4.0 mL of a 50 mM sodium phosphate buffer (PBS, pH 7.0), and centrifuged at 19,934× *g* at 4 °C for 20 min for peroxidase (POD) and polyphenol oxidase (PPO) activity analysis. POD activity was measured using 100 µL of the sample mixed with 2.9 mL of a reaction mixture prepared with PBS pH 7.0, 0.05% guaiacol, and 10 mM H_2_O_2_ [42]. The mixture was used for the spectrophotometric determination of absorbance at a wavelength of 470 nm at time 0 (t_0_) and after 1 min (t_1_). The results were expressed according to the difference between t_1_ and t_0_. In order to establish the PPO activity, 100 µL of the sample extract were mixed with 1.9 mL of the reaction mix prepared with PBS pH 7.0 and 2.5 mM pyrocatechol [47]. The mixture was subsequently incubated at 25 °C for 30 min and used for spectrophotometric determination at a wavelength of 480 nm [42]. 

To conduct the phenylalanine ammonia lyase (PAL) activity analysis, frozen tissue (0.5 g) taken from each sample was ground in a mortar with liquid N and subsequently mixed thoroughly with 4.0 mL of a 50 mM sodium phosphate buffer (PBS, pH 8.0) and 0.01 g of polyvinylpyrrolidone. The mixture was then centrifuged at 19,934× *g* at 4 °C for 20 min [47,48], and 100 µL of the sample extract were mixed with 1.4 mL of PBS pH 8.0 and 0.5 mL of a 50 mM L-phenylalanine solution. The mixture was incubated at 40 °C for 60 min and used for spectrophotometric determination at a wavelength of 290 nm [42]. All the spectrophotometric analyses were conducted using a Beckman DU^®^-65 spectrophotometer (Beckman Coulter Inc., Fullerton, CA, USA). 

### 2.11. Ascorbic Acid and Dehydroascorbic Acid Contents

Frozen tissue (10.0 g) from each sample was ground in a mortar with liquid N and mixed with 10.0 mL of an extraction solution prepared with 1 L of a 5% methanol/ultra-pure water (*v*/*v*) solution, 0.168 g L^−1^ of sodium fluoride, 21.014 g L^−1^ of citric acid, and 0.50 g L^−1^ of ethylenediaminetetraacetic acid. The mixture was centrifuged at 3823× *g* at 4 °C for 5 min, and the supernatant was filtered. High-performance liquid chromatography (HPLC) analyses were performed using HPLC Series 1200 (Agilent Technologies, Waldbronn, Baden-Württemberg, Germany; column C18 Zorbax Eclipse XDB-C18 4.6 × 150 mm, 5 µm, Agilent Technologies, Santa Clara, CA, USA) after dehydroascorbic acid (DHAA) derivatisation. The derivatisations were performed by mixing 750 µL of sample extract with 250 µL of the o-phenylendiamine (OPDA) aqueous solution and incubating the mixture at 20 °C for 37 min in the dark. The ascorbic acid (AA) and DHAA contents were determined at wavelengths of 261 and 348 nm, respectively [3,14].

### 2.12. Tissue Ion and Salt Contents

Frozen tissue (10.0 g) from each sample was stomached with 10.0 mL of distilled water and then filtered and used for nitrate, phosphate, and calcium carbonate (NO_3_^−^, PO_4_^3−^, and CaCO_3_, respectively) content determination using a refractometric kit (Merck Reflectoquant RQflex2^©^; Merck KGaA, Darmstadt, Hessen, Germany), following the manufacturer’s instructions. 

### 2.13. Sampling Size, Statistical Analysis and Experimental Design

The statistical experimental design was a randomised complete block design (RCBD). A two-factorial experimental design (2 substrate disinfections × 2 *B. subtilis* inoculations × 3 blocks) was adopted. Two factors were considered: (a) the use of disinfected and non-disinfected substrates (DS and NDS, respectively) and (b) the use of inoculated and non-inoculated seeds and substrates (BI and NBI, respectively) obtained by applying *Bacillus subtilis* as PGPR [4]. 

Six bags per treatment per block were used during the postharvest period; we examined one bag per treatment per block at each sampling date throughout the shelf life. The data were subjected to analysis of variance (ANOVA) using the Statistical Package for Social Science (SPSS Version 19.0; SPSS Inc., Chicago, IL, USA). The interactions were represented by the standard error.

## 3. Results and Discussion

Beneficial microorganisms could be considered a preharvest biotic factor that affects fruit and vegetable quality and enhances shelf lives [49,50]. In this study, the phytochemical content of fresh-cut watercress, grown with or without *Bacillus subtilis,* during the shelf-life period was evaluated. The results showed that the O_2_ content in the headspace of all the fresh-cut salad bags was consumed, CO_2_ was produced [23,51,52,53], and the respiration rate was intense in the first five days of the shelf life (Table 1). 

In particular, the *B. subtilis* inoculation increased the O_2_ content at d1 and d7 (Table 1). BI had a higher O_2_ content at d1 and CO_2_ at d5 than NBI (+2% and +4%, respectively). DS had a higher CO_2_ concentration at d5 and O_2_ content at d11 than NDS (+24% and +82%, respectively). The average CO_2_ content of the bag headspace at d11 was ca. 8.5%. Additionally, the substrate disinfection augmented these two parameters over time, except for the CO_2_ concentration at d11. NDS had a higher O_2_ content and lower CO_2_ concentration than DS (+5% and −18%, respectively) at d1. The disinfection × *B. subtilis* interaction decreased the O_2_ and CO_2_ amount in the headspace of the fresh-cut salad bags from d3 to d9, with the exception of d5 for the CO_2_ content (Table 1). 

DS × BI had the lowest O_2_ content at d3 and d5 and the highest CO_2_ content at d3. NDS × BI showed the highest O_2_ content and the lowest CO_2_ content in bag headspace at d3 and d7. NDS × NBI had the highest O_2_ content at d5 and d9 and the lowest CO_2_ content at d9. DS × NBI had the lowest O_2_ amount and the highest CO_2_ content at both d7 and d9. The CO_2_ concentration at levels above 10% in bags during refrigerated storage was due to the high permeance of the film used and the reduced cutting surface of plants, which slowed the respiration rate and extended the shelf life [22]. The high O_2_ and low CO_2_ values were also reported in a paper from 2014 [40] in which the authors studied the effects of growing cycle and nutrient solution aeration on the yield, quality, and on shelf life of watercress grown in a floating system. 

During storage, the FWL was not affected by the treatments (data not shown) and was less than 1% of the original weight at the end of the shelf life. The upper threshold limit value to consider fresh leafy green vegetables marketable is 2% [26]. These results are in line with other studies on BLVs such as garden cress (*Lepidium sativum* L.) [42,54].

The disinfection × *B. subtilis* interaction determined a higher accumulation of chlorophylls at d3 and d7 (Table 2). Specifically, DS × BI had the highest chlorophyll content at d3 and d7; the lowest chlorophyll contents at d3 and d7 were found in NDS × BI and DS × NBI, respectively. At d9, DS × NBI had the highest *Car*. content and NDS × BI had the lowest *Car*. content (Table 3).

These results were confirmed by other studies in which treatment with biostimulant products influenced the functioning of enzymes in the photosynthetic process, causing the stimulation of Ribulose-1,5- bisphosphate carboxylase/oxygenase [55,56,57]. 

Decreases in chlorophyll content during shelf life have already been described in many species, e.g., for baby leaf lettuce (*Lactuca sativa* L.) [58] and cultivated rocket (*Eruca sativa* Mill.) grown in FGS [59]. The chlorophyll degradative evolution during the shelf life has shown a different trend: the Chl. *a* content reduction was faster than that of Chl. *b* for frozen watercress during storage [42,60]. The higher amount of Chl. *a* than Chl. *b* measured over time could be caused by the transformation of Chl. *b* into Chl. *a* prior to its degradation [8,9]. 

All photosynthetic organisms accumulate carotenoids that play essential roles in photosystem assembly, photoprotection, and light harvesting. Usually, plant tissues accumulate lutein, *b*-carotene, violaxanthin and neoxanthin: the changes in the composition of these compounds alter photosynthesis and photoprotection [61]. In our study, the *Car.* content detected in the fresh-cut watercress remained constant during the entire shelf-life period. Similar data have been reported for β-carotene in spinach (*Spinacia oleracea* L.), lamb’s lettuce (*Valerianella locusta* L.), and mizuna (*Brassica rapa* L.). On the contrary, small to moderate decreases have been observed in watercress during shelf life [62]

Studies on antioxidant activity of vegetables have been commonly limited to phenols and flavonoids, as well as to the influence of single phenolics on antioxidant activity, but they have often forgotten the importance of antioxidant activity. Additionally, most studies have been conducted on plants cultivated under conventional farming [63]. Therefore, results from cultivation in soilless systems can be valuable. Here, NBI had a higher antioxidant capacity at d1 than BI (+8%), while BI had a higher AC than NBI at d3 and d9 (+6% and +12%, respectively). In general, AC was steady throughout the entire fresh-cut watercress shelf life, as already observed in fresh-cut garden cress stored in a light condition at 4 °C for 5 d [42] and fresh-cut baby leaf spinach stored at 5 °C for 10 d [23]. In our study, DS had a higher AC than NDS (ca. +19%) at d1, d3, d9, and d11. Both the main factors affected the AC over time with the exception of the *B. subtilis* inoculation at d11 (Table 4). The disinfection × *B. subtilis* interaction increased the AC at d5 and d7 (Table 4). DS × BI at d5 and DS × NBI at d7 showed the highest AC. The lowest AC was found in both samplings in NDS × NBI. Similar results were obtained in a recent study in which the addition of compost extracts to a nutrient solution improved the content of potentially healthy substances compounds such as total phenols and flavonoids, as well as antioxidant capacity, and reduced the senescence of plant tissues trough the production of 1-aminocyclopropane-1-carboxylase [64].

BI showed a higher So-Q content than NBI at both d7 and d11 (+26% and +11%, respectively) (Table 5). At d5, treatments had no significant effect on the So-Q content, the average value of which was 0.48 Abs_437_. The So-Q content increased in an approximatively constant fashion from d1 to d11, except at d5 when a peak was registered. The So-Q content measured in the leafy tissue is the result of the combination of the browning enzyme activities, the AA content (which is responsible for the PPO catalytic activity inhibitions), and the TP oxidation to quinones [65]. NDS × BI had the lowest BP at d5 and the highest So-Q content at d3. DS had a higher BP and So-Q contents at d7 than NDS (+23% and +27%, respectively). The BP decrease in fresh-cut watercress observed in the present experiment could have been due to the storage temperature used for the shelf-life simulation, which slows phenolic compound oxidation [66,67]. 

One reaction to stress is the production of reactive oxygen species (ROS) that trigger oxidative stress [68]. Plants control ROS by enzymatic antioxidants, including peroxidase [69]. In our experiment, the substrate disinfection decreased POD enzyme activities over time, except for at d3 (Table 6). The *B. subtilis* inoculation decreased the POD activity at both d1 and d5 and increased the PPO activity at d7. PAL activity diminished at d1, d3, and d9 (Table 6 and Table 7). The disinfection × *B. subtilis* interaction significantly influenced the POD, PPO and PAL activities over time, with the exception of the POD activity at d3 and the PAL activity at both d3 and d9 (Table 6 and Table 7). At d1, NDS × NBI had the highest POD activity and NDS × BI had the lowest activity. DS × NBI had the highest POD activity at both d5 and d7; DS × BI had the lowest POD activity from d5 to d11. During the shelf life of the last two samplings, the highest POD activity was found in NDS × BI. NDS had a higher POD activity than DS at d3 (+13%). NDS × BI had the highest PPO activity at d1, d5 and d11. DS × BI had the lowest PPO activity at d1, d9 and d11 and the highest PPO activity at d7. DS × NBI showed the highest PPO activity at d3, and NDS × NBI showed the lowest values from d3 to d7. NDS × NBI also had the highest PPO activity at d9. DS × NBI had the highest PAL activity at both d1 and d11. 

The lowest PAL activity was found in NDS × NBI at d1, while the lowest value was found in DS × BI at d11. The highest PAL activity was found at d5 and d7 in DS × BI. The lowest PAL activity was observed in DS × NBI at d5 and in NDS × BI at d7. At both d3 and d9, DS had a higher PAL activity than NDS (+24% and +13%, respectively) and NBI had a higher PAL activity than BI (+11% and +10%, respectively). The POD, PPO and PAL activity increased during the shelf-life period. Specifically, the trend was linear for the POD and PPO and variable for PAL throughout the postharvest period. Improvements in the enzyme activities during shelf life have already been reported for fresh-cut garden cress stored in a light condition at 4 °C for 5 d [42], for fresh-cut cultivated rocket stored at 4 °C for 10 d [59], and for fresh-cut fully mature head Romaine lettuce subject to high-intensity light (HIL, 2500 lx), low-intensity light (LIL, 500 lx), darkness, and storage at 4 °C for 7 d [65]. 

Increased PAL activity and related enzymes that function in secondary metabolism has been found in a few species [69,70] after the supply of biostimulants, such as PGPR [71]. 

Vegetables are rich in various phytochemicals, and biologically active substances with beneficial health effects such as ascorbic acid [72]. Here, the substrate disinfection significantly influenced the AA content at both d3 and d5; both the main factors affected the DHAA content over time, apart from at d9 of the shelf life (Table 8). The disinfection × *B. subtilis* interaction decreased the ascorbic acid content at d1, d5 and d9 and the DHAA content at d9 (Table 8). NDS × NBI had the highest AA content at d1 and the highest DHAA content at d9. At the same samplings, the lowest AA and DHAA contents were found in DS × NBI. The highest and lowest AA contents were found in NDS × BI and DS × BI, respectively, at d5. 

Opposite results were found at d9. At d3, DS × BI and DS × NBI had the highest and lowest DHAA contents, respectively. The watercress grown in NDS at d9 had a higher AA content than that grown in DS (+56%). The AA content was below the detection limit at both d7 and d11. The DHAA content was higher in DS than in NDS at d1, d7 and d11 (ca. +24%), and NDS had a higher AA content than DS at d5 (+18%). NBI had a higher DHAA content than BI at both d1 and d5 (+8% and +23%, respectively), which contrasted what occurred at d7 and d11 when BI had a higher DHAA content than NBI (+16% and +63%, respectively).

In this study, the AA-to-DHAA ratio remained constantly in favour of DHAA during the entire shelf-life period. The general reduction in vitamin C registered for fresh-cut watercress could have been due to the absence of light in the display cabinet, as light is one of the factors responsible for new AA synthesis [73]. 

Regarding the mineral nutrient content, the PO_4_^3−^ content was higher in BI than in NBI at d1 (+10%). The disinfection × *B. subtilis* interaction showed an increased value for the PO_4_^3−^ content at d3, d9 and d11; both of the main factors influenced the PO_4_^3−^ content over time, with the exception of the *B. subtilis* inoculation at d5 (Table 9). DS × BI showed the highest PO_4_^3−^ content at d3 and d11 and the lowest content at d7 and d9. NDS × BI had the lowest PO_4_^3−^ content at d3 and the highest content at d9. NDS × NBI had the highest PO_4_^3−^ content at d7. DS × NBI had the lowest value at d11. DS had a higher PO_4_^3−^ content than NDS at both d1 and d5 (ca. +16%). 

The substrate disinfection increased the CaCO_3_ content over time, apart from at d3; the *B. subtilis* inoculation increased the CaCO_3_ content from d1 to d5 and at d11 (Table 9). The disinfection × *B. subtilis* interaction decreased the CaCO_3_ content at d3, d5 and d9; NDS × NBI had the highest CaCO_3_ content at d3 and d9; NDS × BI had the lowest CaCO_3_ content at d3 and the highest at d5; and DS × NBI had the lowest CaCO_3_ content at both d5 and d9. NDS resulted in a higher CaCO_3_ content than DS at d1, d7 and d11 (ca. +35%). NBI showed a higher CaCO_3_ content at d1 than BI (+8%), which had a higher CaCO_3_ content than NBI at d11 (+15%). 

Nitrate content is an important quality characteristic, and the EU encourages good agricultural practices to reduce the nitrate contents in leafy vegetables [74]. In this study, the substrate disinfection decreased the NO_3_^−^ content over time; the *B. subtilis* inoculation affected the NO_3_^−^ content at d1 and from d5 to d9 (Table 10). The interaction between disinfection × *B. subtilis* evidenced a reduction in NO_3_^−^ content at d1, d3, d9 and d11; NDS × NBI had the highest NO_3_^−^ content at d1, and the lowest content was found in DS × BI, not only at d1 but also at d3 and d11. NDS × BI showed the highest NO_3_^−^ content at d3, d9 and d11. The lowest NO_3_^−^ content at d9 was found in DS × NBI. NDS and NBI had a higher NO_3_^−^ content than DS and BI at both d5 and d7 (ca. +81% and ca. +12%, respectively). In general, the ion content in fresh-cut watercress was stable or slowly decreased during the shelf life, as also reported in other studies [75,76] in which the authors demonstrated that that the nitrate content remained constant in different types of green lettuce during storage. Specifically, Konstantopoulou and co-authors [76] found that the nitrate level in lettuce decreased at storage temperatures of 5 and 10 °C for 10 days. Siomos et al. [77] reported that the nitrate content found in romaine lettuce remained unchanged during storage at 1 °C from 3 to 15 days. Similar results were obtained in a study in which the authors indicated that the nitrate content of spinach was unchanged during storage at 5 °C for 7 days but was reduced at higher temperatures [78].

## 4. Conclusions

Watercress is a rarely studied BLV that is suitable for processing as a fresh-cut product. During shelf life in this study, the watercress respiration rate followed the expected O_2_ consumption and CO_2_ production trends in the fresh-cut salad bag headspace, with the former only being below 2% at d11. The inherent phytochemical compounds were mostly retained in fresh-cut salad bags prepared with watercress grown in a disinfected substrate throughout the shelf-life period but less clearly in the seeds and substrates inoculated with PGPR. The external quality, during the shelf life, was slightly affected by a pigment content decrease and a browning enzyme activity increase. These metabolic reactions occurred at different intensities according to the different watercress growing conditions. The effect of *Bacillus subtilis* during the postharvest period was not fully exploited due to inconsistency in its performance during our study. The results evidenced the complex and cumulative effects of *Bacillus subtilis* due to the interactions between plants, microorganism, and environmental factors. The mechanisms adopted by these rhizobacteria in the postharvest physiology of plants remain to be fully explored. Indeed, different plant species or cultivars can produce different types of root exudates that support the activity of the inoculated microorganisms and also serve as substrates for the formation of biologically active substances by the microorganisms. The inoculated microorganisms must survive in the selected formulation and produce the desired activity following inoculation in the field. Additional research is thus needed to gain a better understanding of the fate of the phytochemical content in fresh-cut watercress treated with PGPR during the shelf-life period, particularly in function of the raw material quality. 

## Figures and Tables

**Table 1 plants-11-00589-t001:** Bag headspace gas composition (O_2_ and CO_2_) in the fresh-cut watercress during its shelf life (NDS: non-disinfected substrate; DS: disinfected substrate; NBI: non-inoculated seeds and substrates; BI: seeds and substrates inoculated with *B. subtilis*).

	O_2_ (%)										CO_2_ (%)												
	d1		d3		d5		d7		d9		d11	d1		d3		d5		d7		d9		d11	
Disinfection																							
NDS	15.70	a ^z^	10.78	a	7.08	a	8.08	a	4.17	a	0.90	b	3.28	b	6.55	b	8.12	b	7.20	b	8.43	b	8.52
DS	14.90	b	8.75	b	2.97	b	5.67	b	2.70	b	1.64	a	3.98	a	7.98	a	10.03	a	8.70	a	9.35	a	8.52
*B. subtilis*																							
NBI	15.18	b	9.72		5.55	a	6.30	a	4.38		1.10		3.70		7.30		8.92	b	8.17	a	8.38		8.72
BI	15.42	a	9.82		4.50	b	7.45	b	3.05		1.44		3.57		7.23		9.23	a	7.73	b	9.08		8.32
Disinfection × *B. subtilis*																							
NDS × NBI	15.63		10.13		7.27		7.07		4.97		0.73		3.30		6.97		8.07		7.57		8.03		8.80
NDS × BI	15.77		11.43		6.90		9.10		3.37		1.15		3.27		6.13		8.17		6.83		8.83		8.10
DS × NBI	14.73		9.30		3.83		5.53		2.60		1.65		4.10		7.63		9.77		8.77		9.40		8.60
DS × BI	15.07		8.20		2.10		5.80		2.73		1.63		3.87		8.33		10.30		8.63		9.33		8.47
Mean	15.30		9.77		5.03		6.88		3.58		1.27		3.63		7.27		9.08		7.95		8.80		8.52
SE	0.09		0.17		0.25		0.17		0.46		0.29		0.07		0.08		0.13		0.06		0.17		0.09
Significance																							
Disinfection	<0.001		<0.001		<0.001		<0.001		0.001		0.023		<0.001		<0.001		<0.001		<0.001		<0.001		0.177
*B. subtilis*	0.014		0.559		<0.001		<0.001		0.807		0.797		0.071		0.377		0.020		<0.001		0.314		0.054
Disinfection × *B. subtilis*	0.286		<0.001		0.007		<0.001		0.016		0.749		0.174		<0.001		0.108		<0.001		0.008		0.068

^z^ Values in the same column following the same letter are not statistically different at *p* < 0.05 according to F-test. Disinfection × *B. subtilis* values are the means of the replicates. SE: standard error.

**Table 2 plants-11-00589-t002:** Chlorophyll *a* (Chl. *a*) and chlorophyll *b* (Chl. *b*) contents in the fresh-cut watercress during its shelf life (NDS: non-disinfected substrate; DS: disinfected substrate; NBI: non-inoculated seeds and substrates; BI: seeds and substrates inoculated with B. subtilis). The results are expressed per fresh weight (FW).

	Chl. *a* (mg g^−1^ FW)												Chl. *b* (mg g^−1^ FW)											
	d1		d3		d5		d7		d9		d11		d1		d3		d5		d7		d9		d11	
Disinfection																								
NDS	0.52	b ^z^	0.42	b	0.37	b	0.52		0.41	b	0.29	b	0.15	b	0.11	b	0.10	b	0.15		0.10	b	0.08	b
DS	0.74	a	0.55	a	0.51	a	0.52		0.51	a	0.46	a	0.22	a	0.16	a	0.14	a	0.15		0.13	a	0.12	a
*B. subtilis*																								
NBI	0.63		0.43	b	0.44		0.51		0.53	a	0.41	a	0.18		0.12	b	0.12		0.15	b	0.13	a	0.12	a
BI	0.63		0.54	a	0.44		0.53		0.38	b	0.33	b	0.18		0.16	a	0.12		0.16	a	0.09	b	0.09	b
Disinfection × *B. subtilis*																								
NDS × NBI	0.51		0.43		0.38		0.55		0.47		0.32		0.15		0.12		0.10		0.16		0.12		0.09	
NDS × BI	0.54		0.41		0.36		0.49		0.34		0.25		0.15		0.11		0.09		0.15		0.08		0.07	
DS × NBI	0.74		0.43		0.50		0.48		0.59		0.49		0.22		0.13		0.13		0.14		0.15		0.14	
DS × BI	0.73		0.67		0.53		0.57		0.42		0.42		0.21		0.20		0.14		0.17		0.10		0.11	
Mean	0.63		0.48		0.44		0.52		0.46		0.37		0.18		0.14		0.12		0.15		0.11		0.10	
SE	0.02		0.02		0.02		0.01		0.02		0.01		0.01		0.01		0.01		<0.01		<0.01		<0.01	
Significance																								
Disinfection	<0.001		<0.001		<0.001		0.712		<0.001		<0.001		<0.001		<0.001		<0.001		0.253		<0.001		<0.001	
*B. subtilis*	0.809		<0.001		0.957		0.094		<0.001		<0.001		0.817		<0.001		0.707		0.003		<0.001		<0.001	
Disinfection × *B. subtilis*	0.454		<0.001		0.208		<0.001		0.229		0.906		0.448		<0.001		0.210		<0.001		0.212		0.229	

^z^ Values in the same column following the same letter are not statistically different at *p* < 0.05 according to F-test. Disinfection × *B. subtilis* values are the means of the replicates. SE: standard error.

**Table 3 plants-11-00589-t003:** Carotenoid (*Car.*) content in the fresh-cut watercress during its shelf life (NDS: non-disinfected substrate; DS: disinfected).

	*Car.* (mg g^−1^ FW)											
	d1		d3		d5		d7		d9		d11	
Disinfection												
NDS	0.19	b^z^	0.08	b	0.15	b	0.20		0.17	b	0.14	b
DS	0.27	a	0.13	a	0.21	a	0.20		0.21	a	0.19	a
*B. subtilis*												
NBI	0.23		0.08	b	0.18		0.19		0.22	a	0.18	a
BI	0.23		0.12	a	0.18		0.20		0.16	b	0.15	b
Disinfection × *B. subtilis*												
NDS × NBI	0.19		0.08		0.16		0.21		0.19		0.15	
NDS × BI	0.20		0.07		0.15		0.19		0.14		0.12	
DS × NBI	0.27		0.08		0.20		0.18		0.24		0.21	
DS × BI	0.26		0.17		0.21		0.21		0.17		0.18	
Mean	0.23		0.10		0.18		0.20		0.19		0.17	
SE	0.01		0.01		0.01		<0.01		0.01		0.01	
Significance												
Disinfection	<0.001		<0.001		<0.001		0.300		<0.001		<0.001	
*B. subtilis*	0.965		<0.001		0.913		0.091		<0.001		<0.001	
Disinfection × *B. subtilis*	0.305		<0.001		0.169		<0.001		0.009		0.491	

^z^ Values in the same column following the same letter are not statistically different at *p* < 0.05 according to F-test. Disinfection × *B. subtilis* values are the means of the replicates. SE: standard error.

**Table 4 plants-11-00589-t004:** Antioxidant capacity (AC) and total phenolic (TP) content in the fresh-cut watercress during its shelf life (NDS: non-disinfected substrate; DS: disinfected substrate; NBI: non-inoculated seeds and substrates; BI: seeds and substrates inoculated with B. subtilis). The results are expressed per fresh weight (FW).

	AC (μmol Fe^2+^ g^−1^ FW)												TP (mg Gallic Acid g^−1^ FW)											
	d1		d3		d5		d7		d9		d11		d1		d3		d5		d7		d9		d11	
Disinfection																								
NDS	27.57	b ^z^	28.70	b	27.84	b	31.79	b	27.45	b	28.84	b	0.73	b	0.81	b	0.62	b	0.57	b	0.65	b	0.67	b
DS	32.07	a	33.75	a	36.71	a	36.75	a	34.61	a	32.87	a	0.92	a	0.95	a	0.75	a	0.72	a	0.80	a	0.72	a
*B. subtilis*																								
NBI	30.95	a	30.38	b	31.39	b	35.17	a	29.20	b	31.05		0.85	a	0.88		0.69		0.69	a	0.71		0.70	
BI	28.68	b	32.07	a	33.16	a	33.36	b	32.85	a	30.65		0.81	b	0.89		0.69		0.61	b	0.74		0.69	
Disinfection × *B. subtilis*																								
NDS × NBI	28.25		27.40		26.13		31.69		25.04		29.15		0.76		0.81		0.59		0.62		0.61		0.67	
NDS × BI	26.89		30.00		29.55		31.89		29.85		28.52		0.71		0.82		0.66		0.53		0.69		0.68	
DS × NBI	33.65		33.36		36.65		38.65		33.37		32.95		0.93		0.95		0.79		0.76		0.81		0.74	
DS × BI	30.48		34.15		36.77		34.84		35.85		32.79		0.91		0.95		0.72		0.68		0.80		0.70	
Mean	29.82		31.23		32.28		34.27		31.03		30.85		0.83		0.88		0.69		0.65		0.73		0.69	
SE	0.70		0.53		0.72		0.57		0.97		0.41		0.01		0.02		0.01		0.01		0.02		0.01	
Significance																								
Disinfection	<0.001		<0.001		<0.001		<0.001		<0.001		<0.001		<0.001		<0.001		<0.001		<0.001		<0.001		0.003	
*B. subtilis*	0.002		0.002		0.016		0.002		<0.001		0.332		<0.001		0.580		0.836		<0.001		0.055		0.224	
Disinfection × *B. subtilis*	0.200		0.093		0.024		0.001		0.231		0.571		0.179		0.795		<0.001		0.940		0.018		0.068	

^z^ Values in the same column following the same letter are not statistically different at *p* < 0.05 according to F-test. Disinfection × *B. subtilis* values are the means of the replicates. SE: standard error.

**Table 5 plants-11-00589-t005:** Browning potential (BP) and soluble o-quinone (So-Q) content in the fresh-cut watercress during its shelf life (NDS: non-disinfected substrate; DS: disinfected substrate; NBI: non-inoculated seeds and substrates; BI: seeds and substrates inoculated with *B. subtilis*).

	BP (Abs_340_)												So-Q (Abs_437_)											
	d1		d3		d5		d7		d9		d11		d1		d3		d5		d7		d9		d11	
Disinfection																								
NDS	1.09	b ^z^	0.99	b	1.10	b	0.90	b	0.95		0.98	b	0.24	b	0.33		0.47		0.20	b	0.20	b	0.37	
DS	1.17	a	1.12	a	1.15	a	1.11	a	0.97		1.07	a	0.29	a	0.30		0.49		0.26	a	0.24	a	0.38	
*B. subtilis*																								
NBI	1.19	a	1.07		1.11		0.99		1.02	a	1.03		0.27		0.29	b	0.49		0.21	b	0.22		0.36	b
BI	1.07	b	1.04		1.14		1.02		0.90	b	1.02		0.26		0.34	a	0.47		0.26	a	0.22		0.40	a
Disinfection × *B. subtilis*																								
NDS × NBI	1.09		0.96		1.12		0.90		0.97		0.94		0.22		0.29		0.48		0.17		0.20		0.35	
NDS × BI	1.10		1.01		1.08		0.91		0.93		1.02		0.26		0.37		0.45		0.24		0.21		0.39	
DS × NBI	1.30		1.18		1.10		1.08		1.07		1.12		0.31		0.30		0.49		0.24		0.24		0.36	
DS × BI	1.05		1.07		1.20		1.14		0.88		1.02		0.26		0.31		0.48		0.28		0.24		0.41	
Mean	1.13		1.06		1.12		1.01		0.96		1.02		0.26		0.32		0.48		0.23		0.22		0.38	
SE	0.04		0.02		0.02		0.03		0.02		0.03		0.01		0.02		0.01		0.01		<0.01		0.01	
Significance																								
Disinfection	0.023		<0.001		0.041		<0.001		0.235		0.003		<0.001		0.091		0.093		<0.001		<0.001		0.269	
*B. subtilis*	0.001		0.138		0.261		0.291		<0.001		0.632		0.814		0.010		0.098		<0.001		0.390		0.004	
Disinfection × *B. subtilis*	<0.001		<0.001		0.006		0.401		<0.001		0.003		<0.001		0.038		0.316		0.116		0.022		0.656	

^z^ Values in the same column following the same letter are not statistically different at *p* < 0.05 according to F-test. Disinfection × *B. subtilis* values are the means of the replicates. SE: standard error.

**Table 6 plants-11-00589-t006:** Peroxidase (POD) and polyphenol oxidase (PPO) activities in the fresh-cut watercress during its shelf life (NDS: non-disinfected substrate; DS: disinfected substrate; NBI: non-inoculated seeds and substrates; BI: seeds and substrates inoculated with *B. subtilis*). The results are expressed per fresh weight (FW).

	POD (Unit g^−1^ FW)												PPO (Unit g^−1^ FW)											
	d1		d3		d5		d7		d9		d11		d1		d3		d5		d7		d9		d11	
Disinfection																								
NDS	2.36	b ^z^	3.16	a	3.53	a	3.24		4.25	a	5.18	a	11.87		14.26	b	19.61		19.10	b	32.14	a	31.64	
DS	2.55	a	2.80	b	3.13	b	3.19		3.31	b	4.31	b	11.33		14.63	a	19.46		21.13	a	30.33	b	31.13	
*B. subtilis*																								
NBI	2.58	a	2.94		3.83	a	3.28		3.51		4.52		11.80		14.32		19.32		18.96	b	33.09	a	31.20	
BI	2.34	b	3.03		2.84	b	3.16		4.05		4.98		11.39		14.57		19.75		21.28	a	29.39	b	31.57	
Disinfection × *B. subtilis*																								
NDS × NBI	2.61		3.14		3.71		3.16		3.13		4.38		11.12		13.83		19.10		18.38		33.45		30.26	
NDS × BI	2.12		3.18		3.35		3.32		5.36		5.99		12.61		14.69		20.12		19.83		30.84		33.01	
DS × NBI	2.55		2.74		3.95		3.40		3.88		4.65		12.47		14.81		19.54		19.54		32.72		32.14	
DS × BI	2.56		2.87		2.32		2.99		2.73		3.97		10.18		14.44		19.39		22.72		27.94		30.12	
Mean	2.46		2.98		3.33		3.22		3.78		4.75		11.60		14.44		19.54		20.12		31.24		31.38	
SE	0.05		0.07		0.14		0.08		0.42		0.25		0.45		0.18		0.29		0.39		0.43		0.61	
Significance																								
Disinfection	<0.001		<0.001		0.006		0.509		0.029		0.001		0.234		0.039		0.614		<0.001		<0.001		0.411	
*B. subtilis*	<0.001		0.228		<0.001		0.118		0.208		0.063		0.371		0.165		0.133		<0.001		<0.001		0.556	
Disinfection × *B. subtilis*	<0.001		0.584		<0.001		<0.001		<0.001		<0.001		<0.001		0.001		0.046		0.027		0.014		<0.001	

^z^ Values in the same column following the same letter are not statistically different at *p* < 0.05 according to F-test. Disinfection × *B. subtilis* values are the means of the replicates. SE: standard error.

**Table 7 plants-11-00589-t007:** Phenylalanine ammonia lyase (PAL) activity in the fresh-cut watercress during its shelf life (NDS: non-disinfected substrate; DS: disinfected substrate; NBI: non-inoculated seeds and substrates; BI: seeds and substrates inoculated with *B. subtilis*). The results are expressed per fresh weight (FW).

	PAL (µmol cinnamic acid h^−1^ g^−1^ FW)											
	d1		d3		d5		d7		d9		d11	
Disinfection												
NDS	0.102	b ^z^	0.119	b	0.129		0.103	b	0.151	b	0.154	
DS	0.115	a	0.148	a	0.132		0.118	a	0.171	a	0.149	
*B. subtilis*												
NBI	0.113	a	0.140	a	0.127		0.105	b	0.169	a	0.155	
BI	0.103	b	0.126	b	0.134		0.116	a	0.154	b	0.148	
Disinfection × *B. subtilis*												
NDS × NBI	0.088		0.125		0.138		0.109		0.156		0.150	
NDS × BI	0.116		0.113		0.120		0.097		0.146		0.158	
DS × NBI	0.138		0.155		0.116		0.102		0.182		0.159	
DS × BI	0.091		0.140		0.148		0.135		0.161		0.138	
Mean	0.108		0.133		0.130		0.110		0.161		0.151	
SE	0.004		0.003		0.004		0.003		0.003		0.006	
Significance												
Disinfection	*0.005*		*<0.001*		0.465		*<0.001*		*<0.001*		0.451	
*B. subtilis*	*0.023*		*<0.001*		0.133		*<0.001*		*<0.001*		0.302	
Disinfection × *B. subtilis*	*<0.001*		0.656		*<0.001*		*<0.001*		0.115		*0.026*	

^z^ Values in the same column following the same letter are not statistically different at *p* < 0.05 according to F-test. Disinfection × *B. subtilis* values are the means of the replicates. SE: standard error.

**Table 8 plants-11-00589-t008:** Ascorbic acid (AA) and dehydroascorbic acid (DHAA) contents in the fresh-cut watercress during its shelf life (NDS: non-disinfected substrate; DS: disinfected substrate; NBI: non-inoculated seeds and substrates; BI: seeds and substrates inoculated with *B. subtilis*). The results are expressed per fresh weight (FW).

	AA (mg g^−1^ FW)												DHAA (mg g^−1^ FW)											
	d1		d3		d5		d7		d9		d11		d1		d3		d5		d7		d9		d11	
Disinfection																								
NDS	0.38	^z^	0.23	a	0.19	a	-		0.14		-		14.59	b	10.89	b	7.08	a	6.88	b	4.28		1.90	b
DS	0.35		0.15	b	0.14	b	-		0.14		-		17.01	a	11.09	a	5.98	b	8.81	a	4.24		2.40	a
*B. subtilis*																								
NBI	0.37		0.19		0.16		-		0.14		-		16.39	a	10.32	b	7.20	a	7.27	b	4.26		1.64	b
BI	0.37		0.18		0.17		-		0.14		-		15.21	b	11.66	a	5.86	b	8.43	a	4.26		2.66	a
Disinfection × *B. subtilis*							-				-													
NDS × NBI	0.43		0.23		0.14		-		0.17		-		15.26		11.25		7.63		6.29		4.58		1.31	
NDS × BI	0.34		0.23		0.24		-		0.10		-		13.92		10.53		6.52		7.48		3.98		2.50	
DS × NBI	0.30		0.15		0.17		-		0.11		-		17.53		9.39		6.77		8.24		3.95		1.97	
DS × BI	0.40		0.14		0.10		-		0.18		-		16.49		12.78		5.19		9.39		4.54		2.83	
Mean	0.37		0.19		0.16		-		0.14		-		15.80		10.99		6.53		7.85		4.26		2.15	
SE	0.03		0.01		0.01		-		0.01		-		0.35		0.07		0.28		0.20		0.10		0.15	
Significance																								
Disinfection	0.277		<0.001		<0.001		-		0.761		-		<0.001		0.009		<0.001		<0.001		0.735		0.001	
*B. subtilis*	0.982		0.846		0.122		-		0.934		-		0.001		<0.001		<0.001		<0.001		0.954		<0.001	
Disinfection × *B. subtilis*	0.002		0.855		<0.001		-		<0.001		-		0.673		<0.001		0.404		0.928		<0.001		0.265	

^z^ Values in the same column following the same letter are not statistically different at *p* < 0.05 according to F-test. Disinfection × *B. subtilis* values are the means of the replicates. SE: standard error.

**Table 9 plants-11-00589-t009:** Phosphate (PO_4_^3−^) and calcium carbonate (CaCO_3_) contents in the fresh-cut watercress during its shelf life (NDS: non-disinfected substrate; DS: disinfected substrate; NBI: non-inoculated seeds and substrates; BI: seeds and substrates inoculated with *B. subtilis*). The results are expressed per fresh weight (FW).

	PO_4_^3−^ (mg g^−1^ FW)												CaCO_3_ (mg g^−1^ FW)											
	d1		d3		d5		d7		d9		d11		d1		d3		d5		d7		d9		d11	
Disinfection																								
NDS	0.120	b ^z^	0.125	b	0.117	b	0.150	a	0.148	a	0.128	b	0.033	a	0.027		0.036	a	0.034	a	0.034	a	0.030	a
DS	0.134	a	0.144	a	0.142	a	0.132	b	0.140	b	0.135	a	0.026	b	0.027		0.024	b	0.026	b	0.024	b	0.021	b
*B. subtilis*																								
NBI	0.121	b	0.130	b	0.129		0.147	a	0.140	b	0.125	b	0.031	a	0.030	a	0.028	b	0.030		0.029		0.024	b
BI	0.133	a	0.138	a	0.130		0.135	b	0.147	a	0.137	a	0.029	b	0.024	b	0.032	a	0.029		0.029		0.027	a
Disinfection × *B. subtilis*																								
NDS × NBI	0.113		0.130		0.118		0.152		0.139		0.126		0.035		0.033		0.033		0.035		0.035		0.028	
NDS × BI	0.128		0.119		0.117		0.149		0.156		0.130		0.032		0.021		0.038		0.032		0.032		0.033	
DS × NBI	0.129		0.130		0.141		0.142		0.141		0.125		0.027		0.027		0.024		0.026		0.024		0.020	
DS × BI	0.139		0.158		0.143		0.122		0.138		0.144		0.025		0.027		0.025		0.026		0.025		0.022	
Mean	0.127		0.134		0.129		0.141		0.144		0.131		0.030		0.027		0.030		0.030		0.029		0.026	
SE	0.003		0.003		0.002		0.004		0.002		0.002		0.001		0.001		<0.001		0.001		<0.001		0.001	
Significance																								
Disinfection	<0.001		<0.001		<0.001		<0.001		0.001		0.005		<0.001		0.884		<0.001		<0.001		<0.001		<0.001	
*B. subtilis*	<0.001		0.001		0.815		0.002		0.003		<0.001		0.012		<0.001		<0.001		0.184		0.208		<0.001	
Disinfection × *B. subtilis*	0.502		<0.001		0.482		0.016		<0.001		0.002		0.505		<0.001		0.002		0.084		<0.001		0.167	

^z^ Values in the same column following the same letter are not statistically different at *p* < 0.05 according to F-test. Disinfection × *B. subtilis* values are the means of the replicates. SE: standard error.

**Table 10 plants-11-00589-t010:** Nitrate (NO_3_^−^) content in the fresh-cut watercress during its shelf life (NDS: non-disinfected substrate; DS: disinfected substrate; NBI: non-inoculated seeds and substrates; BI: seeds and substrates inoculated with *B. subtilis*). The results are expressed per fresh weight (FW).

	NO_3_^−^ (mg g^−1^ FW)											
	d1		d3		d5		d7		d9		d11	
Disinfection												
NDS	2.80	a ^z^	2.17	a	1.96	a	2.11	a	2.27	a	2.14	a
DS	1.62	b	1.43	b	1.09	b	1.16	b	1.39	b	1.16	b
*B. subtilis*												
NBI	2.31	a	1.87		1.64	a	1.69	a	1.75	b	1.66	
BI	2.10	b	1.73		1.41	b	1.58	b	1.91	a	1.64	
Disinfection × *B. subtilis*												
NDS × NBI	2.95		2.11		2.08		2.17		2.13		2.05	
NDS × BI	2.64		2.23		1.84		2.05		2.41		2.24	
DS × NBI	1.67		1.63		1.21		1.21		1.37		1.27	
DS × BI	1.57		1.23		0.97		1.11		1.41		1.05	
Mean	2.21		1.80		1.53		1.64		1.83		1.65	
SE	0.03		0.07		0.04		0.03		0.03		0.04	
Significance												
Disinfection	<0.001		<0.001		<0.001		<0.001		<0.001		<0.001	
*B. subtilis*	<0.001		0.055		<0.001		<0.001		<0.001		0.649	
Disinfection × *B. subtilis*	0.001		0.001		0.930		0.639		<0.001		<0.001	

^z^ Values in the same column following the same letter are not statistically different at *p* < 0.05 according to F-test. Disinfection × *B. subtilis* values are the means of the replicates. SE: standard error.

## Data Availability

All data included in the main text.

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
