# Peer review of "Understanding the Postharvest Phytochemical Composition Fates of Packaged Watercress (Nasturtium officinale R. Br.) Grown in a Floating System and Treated with Bacillus subtilis as PGPR"

_plants, 2022, doi:10.3390/plants11050589_

Round 1

Reviewer 1 Report

Hereafter are reported the comments arising from my revision of the manuscript entitled " Understanding the postharvest phytochemical composition fates of packaged watercress (Nasturtium officinale R. Br.) grown in a floating system and treated with Bacillus subtilis as PGPR".  

I have read at great length the work. I found this article informative in regards to background information.

The manuscript appears to have sufficient scientific quality and may be of interest to the readers of Plant.

The scientific value of the manuscript is good (although not outstanding)

The manuscript brings some new information.

Abstract: good

Introduction; good

Materials and methods: good

Results : good

Discussion: good

Author Response

We thank the reviewer for his positive comments on the quality of the manuscript.

Reviewer 2 Report

the manuscript entitled "Understanding the postharvest phytochemical composition fates of packaged watercress (Nasturtium officinale R. Br.) grown in a floating system and treated with Bacillus subtilis as PGPR" its a very well presented research work with originality and high interst for the reader. 

My only one comment is about the statistics presentation in Tables. For example i suggest to authors to present the comparison of means at interactions (Disinfection * B subtilis) in cases where the significances were <0.005.   

Author Response

We thank the reviewer for the comments. Thank you for pointing out the issue on how best to present the post-hoc mean separations when the interaction is significant for some variables. Instead of presenting the mean separations of each factor at two levels, making the two mean separation a t-test, we thought to be more representative to indicate the SE for all cases, e.g. when interaction was significant (4 days) and when it was not significant was (8 days). We trust this approach is acceptable.

Reviewer 3 Report

Dear authors,

The manuscript presents an interesting work about how the use of biofertilizer could help in postharvest conditions to store product and increase its life. In this way, the manuscript is well written, with adequate structure and description of the methodology, correct exposition of results and discussion.

Only I detected some little mistake

line 129 10^12

line 136 10^8

line146 125-g

Also, in the main title, the use of biofertilizer is one of the big parts of it, but in the document, only is mentioned in the methodology as a tool for treatment, do you know which is the reason for the observed actions? Did you analyze any PGP mechanisms? Are there any other mechanism that explain the good results obtained?

Best regards

Author Response

We thank the reviewer for the positive comments. The answers are given below.

Only I detected some little mistake

line 129 10^12. R: The mistake has been corrected in the text (Line 134).

line 136 10^8. R: The mistake has been corrected in the text (Line 142).

line146 125-g. R: The mistake has been corrected in the text (Line 161).

Also, in the main title, the use of biofertilizer is one of the big parts of it, but in the document, only is mentioned in the methodology as a tool for treatment, do you know which is the reason for the observed actions? Did you analyze any PGP mechanisms? Are there any other mechanism that explain the good results obtained?

R: Over the last 15 years, a great deal of research has been done in the field of biostimulants. The novelty of this manuscript is that the authors have chosen to highlight the results obtained, instead of dwelling on the characteristics of the biostimulant. A very small number of researches have deepened the effects of PGPP during the post-harvest period. Considering the positive results obtained, our research team will plan new experiments to deepen the effects of PGPP during the post-harvest period. The mechanisms activated by biostimulants in plants require in-depth studies and for a long period of time.

Reviewer 4 Report

In the present study, the effect of Bacillus subtilis inoculation on the physiological properties and phytochemical composition of watercress were evaluated. The research work presents an interesting topic and is well organized. However, there are several points to be addressed. 
I have provided some comments and suggestions, which may help authors improve their Manuscript. 

Comments
Line 112. Plants were grown without bacterial inoculation here. Please clarify this part because the following section describes seed inoculation with bacteria.
Line 127. Please describe substrate? Why was only 50% used in the assay?
Line 130: “The seeds were sterilized in 20% NaCl”, why were seeds not sterilized using ethanol or other solutions? How you tested that 20% NaCl is suitable for sterilization, there are salt-tolerant bacteria.
Line 132: “50% of the disinfected substrate (DS) and 50% of the non-disinfected substrate, both inoculated with bacteria? 
“2.2 Bacterial strain and inoculation” section needs improvement. Please clarify treatments that help easily understand all treatments.
The physiological properties of plants after bacterial inoculation were demonstrated well. However Introduction part mentioned about biocontrol ability of PGPR. Lines 88-93. How do bacterial inoculation and disinfection effect post-harvest pathogens? B. subtilis showed a positive effect on watercress's physiological properties and phytochemical composition. What is the behind such beneficial effect? It is also important to study the colonization of bacteria in plant tissue after harvest and their survival. Please add some photos of plant which demonstrate treatment effects.

Author Response

Line 112. Plants were grown without bacterial inoculation here. Please clarify this part because the following section describes seed inoculation with bacteria.

R: We thank the reviewer for the comment. To clarify that some of the plants have been inoculated, while the remaining part has been used as control, the text of the manuscript has been modified (Lines 107-123). We hope that with the changes the text will be clearer.

Line 127. Please describe substrate? Why was only 50% used in the assay?

R: We have added the information required in the text (Lines 129-132).

Line 130. “The seeds were sterilized in 20% NaCl”, why were seeds not sterilized using ethanol or other solutions? How you tested that 20% NaCl is suitable for sterilization, there are salt-tolerant bacteria.

R: We thank the reviewer for the comment and apologize for the typo. We really used NaOCl, not NaCl.

Line 132: “50% of the disinfected substrate (DS) and 50% of the non-disinfected substrate, both inoculated with bacteria?

R: The purpose of the manuscript was also to evaluate the effect of Bacillus subtilis on sterilized or non-sterilized substrate. For this reason, both substrates were inoculated.

“2.2 Bacterial strain and inoculation” section needs improvement. Please clarify treatments that help easily understand all treatments.

The physiological properties of plants after bacterial inoculation were demonstrated well. However Introduction part mentioned about biocontrol ability of PGPR. Lines 88-93. How do bacterial inoculation and disinfection effect post-harvest pathogens? B. subtilis showed a positive effect on watercress's physiological properties and phytochemical composition. What is the behind such beneficial effect? It is also important to study the colonization of bacteria in plant tissue after harvest and their survival.

R: Several species of Bacillus can promote the crop health in different ways. Some of these species directly stimulate plant growth either through enhancement in acquisition of nutrients or through stimulation of the host plant’s defense mechanisms; other species can inhibit or suppress the populations of pathogenic microorganisms and/or pests. Although the distribution, diversity, and population dynamics of this genus have been studied using a variety of techniques, much remains to be learned if we are to improve both basic studies of plant–microbe interactions and bacterial ecology, as well as the efforts to improve agricultural technologies. Unfortunately, the mechanisms adopted by Bacillus subtilis in postharvest physiology of plants remain to be fully investigated.

We thank the reviewer for the interesting and useful comments. The encouraging results obtained in this preliminary study are a starting point to deepen the mechanisms activated by this category of biostimulant, also thanks to the support of colleagues involved in microbiology. The next studies will aim to deepen the physiological mechanisms activated by Bacillus subtilis during the post-harvest period. We are sorry for not being able to show any picture of the plants with visible colonization. Pictures were not taken.

This manuscript is a resubmission of an earlier submission. The following is a list of the peer review reports and author responses from that submission.